# The relationship between night-time light and socioeconomic factors in China and India

Guhuai Han[1], Tao Zhou[2], Yuanheng Sun[1,3]*, Shoujie Zhu[1,4]

**1** Institute of Remote Sensing and Geographic Information Systems, School of Earth and Space Science, Peking University, Beijing, China, **2** School of Business, Northwest Normal University, Lanzhou, China, **3** Environmental Information Institute, Navigation College, Dalian Maritime University, Dalian, China, **4** College of Geoscience and Surveying Engineering, China University of Mining & Technology, Beijing, China

* yhsun@pku.edu.cn

## Abstract

This paper re-examines the relationships between night-time light (NTL) and gross domestic product (GDP), population, road networks, and carbon emissions in China and India. Two treatments are carried out to those factors and NTL, which include simple summation in each administrative region (total data), and summation normalized by region area (density data). A series of univariate regression and multiple regression experiments are conducted in different countries and at different scales, in order to find the changes in the relationship between NTL and every parameter in different situations. Several statistical metrics, such as $R^2$, Mean Relative Error (MRE), multiple regression weight coefficient, and Pearson's correlation coefficient are given special attention. We found that GDP, as a comprehensive indicator, is more representative of NTL when the administrative region is relatively comprehensive or highly developed. However, when these regions are unbalanced or undeveloped, the representation of GDP becomes weak and other factors can have a more important influence on the multiple regression. Differences in the relationship between NTL and GDP in China and India can also be reflected in some other factors. In many cases, regression after normalization with the administrative area has a higher $R^2$ value than the total regression. But it is highly influenced by a few highly developed regions like Beijing in China or Chandigarh in India. After the scale of the administrative region becomes fragmented, it is necessary to adjust the model to make the regression more meaningful. The relationship between NTL and carbon emissions shows obvious difference between China and India, and among provinces and counties in China, which may be caused by the different electric power generation and transmission in China and India. From these results, we can know how the NTL is reflected by GDP and other factors in different situations, and then we can make some adjustments.

## Introduction

In the 1970s, the United States launched the Defense Meteorological Satellite Pro-gram (DMSP). The Operational Lines can System (OLS) carried by the DMSP satellite can capture

**Data Availability Statement:** All relevant data are within the paper and its Supporting Information files.

**Funding:** This work was supported by the General Project of National Social Science Fund of China

(17BJY053). Professor Zhou Tao received a salary from the General Project of National Social Science Fund of China (17BJY053).

**Competing interests:** The authors have declared that no competing interests exist.

faint light radiation on the ground at night, thus producing nighttime light (NTL) images [1]. Compared with various radiation products during the day, NTL has unique advantages in terms of reflecting human socioeconomic activities conditions, so it has been widely used in the analysis of various issues related to socioeconomic, disasters, and energy [2–9]. In recent years, many countries have increased their investment in NTL satellites. The National Polar-Orbiting Partnership's Visible Infrared Imaging Radiometer Suite (NPP/VIIRS) of the United States, Luojia-1 of China, and other satellites have been launched [10–15]. Those NTL products have better spatial, temporal and radiometric resolution compared with DMSP/OLS [10], and have accumulated a large amount of available data [10], which provides sufficient support for the relative applications. Therefore, research related to NTL has expanded significantly over the last 10 years [10–15].

At present, there are three main types of research related to NTL. The first type is related to technical problems such as DMSP/OLS desaturation [16, 17], NPP/VIIRS background noise removal [11], and DMSP/OLS image continuity correction [18]. The second type is to study the relationship between NTL and various parameters [19–24] and analyze their spatial aggregation and temporal variation [25–28]. The third category is to directly use NTL as an indicator instead of socioeconomic parameters in sociology, economics, geography, urban planning and other research fields [29–31] because data on population, Gross Domestic Product (GDP), and energy consumption are difficult to obtain or obtain in time in some countries or regions.

We must have a clear understanding of the essence reflected by NTL, and know the relationship between NTL and various factors before using it as an indicator in various socioeconomic studies. Therefore, it is necessary to make a detailed analysis of how various factors affect NTL. Many studies have discussed the relationship between NTL and various factors, and showed that NTL is significantly related to socioeconomic data such as GDP and population, urban development data such as road network density, urban building area, and energy consumption data such as carbon emissions [2–8, 19–24]. However, most of them only conducted a simple univariate regression, without comprehensive discussion of various elements. Even if there are some multiple regression studies [32], they lack a discussion of the relationships between these parameters and NTL at multiple-scales and for different regions. Therefore, some important phenomena are difficult to explain, such as why India's GDP is far less than China's but the NTL is very bright. In addition, the indicator's standards are inconsistent. Studies sometimes use the total light intensity, sometimes use light area or light intensity density [33, 34], or even define the light index by combining the light area and light intensity with different formulas [25]. However, there is no unified explanation or comparison for the indicators. As such, deeper understanding of NTL is still needed, especially the relationships with many factors in different countries and at different scales.

In this study, we perform regression and multiple regression experiments on NTL and several important economic parameters in China and India at multiple scales. We compare these experimental results to discuss the characteristic change in the relationship between NTL and various elements in different types of regions, and find common reasons for these changes. Through this study, we can deepen our understanding of NTL, and make some reasonable adjustments to NTL data when dealing with complex social and economic problems, so as to improve the persuasiveness of replacing some socioeconomic factors with NTL and the accuracy of the final analysis results.

## Study areas

The study areas of this research are China and India, both of which are large countries in Asia. These two countries have the largest and second-largest population, respectively, in the world.

However, the area of China is ~9.6 million square kilometers and that of India is ~2.98 million square kilometers [35, 36], so their population density is quite different. Moreover, the population distribution is also different because of topographic factors. According to official data from the Statistics Bureau of China and India, China's GDP was more than 30 times that of India's in 2015. Aside from the GDP, the economic structure of the two countries is also quite different. For example, China had fully electrified by the end of 2015 [37], while all villages in India were not electrified until April 2018 [38], as announced by the Indian government. In addition, although China has overall differences in the economic level between its eastern and western regions, every Chinese province is still relatively developed and comprehensive, while the imbalance between Indian States is even more serious [39–41]. Through the experimental data, we find that NTL in India is bright, usually surpassing that of western China and approaching that of eastern China. However, the economic level in India is poorer than that in China. There are 34 provinces in China. Due to the availability of some research data (especially county-level data), our experiments mainly use the data of 31 provinces other than Taiwan Province, Hong Kong and Macau. There are 33 States in India, and we use all of them.

## Data

In this study, we comprehensively analyzed four factors related to NTL [19–24]. See Table 1 for a brief introduction of data. Details of the data are as follows:

### NPP/VIIRS data

The VIIRS sensor has 22 bands, and the subastral point spatial resolution of the Day/Night Band (DNB) is 375 m, so it can identify weak light sources and accurately describe the distribution of ground illumination. Compared with DMSP/OLS data, there is no serious supersaturation problem in VIIRS [42,43], so it is more convenient and reasonable to deal with the more developed cities. VIIRS NTL data after radiation calibration, background noise removal, filtering transient light and cloud removal can be downloaded directly from the VIIRS website (https://eogdata.mines.edu/download_dnb_composites.html) (visited on November 10, 2020). The VIIRS NTL data used in this paper are annual average data from China and India in 2015 (SVDNB_npp_20150101–20151231_75N060E_vcm-orm-ntl_v10_c201701311200.avg_rade9), and the geographical coordinate is D_WGS_1984, as shown in Fig 1. We chose the year of 2015 because it has stable representativeness and the data of various elements in 2015 can also be obtained.

### GDP and population data

China's GDP and population data are from the Resource and Environmental Science Data Center of Chinese Academy of Sciences (www.resdc.cn) (accessed on November 17, 2020).

**Table 1. Introduction to data.**

| Data type | Description | Source of the data |
| --- | --- | --- |
| NPP/VIIRS NTL | Night-time light remote sensing image | https://eogdata.mines.edu/download_dnb_composites.html |
| GDP and Population | GDP and population data of China and India | www.resdc.cn, www.rbi.org.in |
| Road network | Road network length of China and India | www.openstreetmap.org OSM |
| Carbon emission | $CO_2$ emission data of China and India | https://edgar.jrc.ec.europa.eu/ |

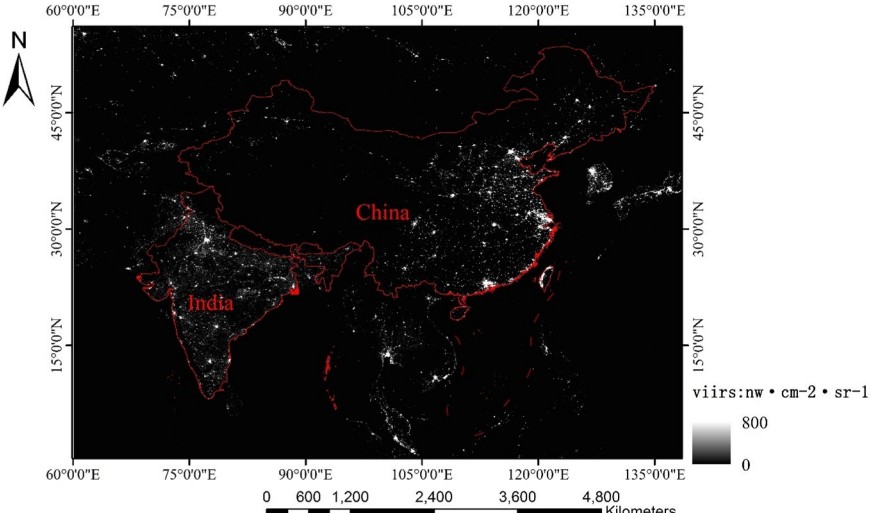

**Fig 1. 2015 annual average VIIRS data in China and India.**

This dataset is based on the national county-level population and GDP statistics, combined with the data of land use and residential land density, and followed by weighted spatialization at a resolution of 1 km.

The GDP and population data of India in 2015 are from Reserve bank of India (www.rbi.org.in) (accessed on November 19, 2020) and Census Commissioner of India (censusindia.gov.in) (accessed on November 19, 2020), respectively. The population data has been interpolated since India's population is surveyed every ten years (the administrative level of state is equivalent to China's province).

### Road network data

The road network data of China and India in 2015 are from OpenStreetMap (OSM, www.openstreetmap.org) (accessed on November 20, 2020). OSM provides infrastructure data such as road network, traffic, land use, and buildings in all countries of the world in Shapefile format. The building data mainly include important landmark buildings, while the road network data are relatively comprehensive. We think the road network can reflect both urban construction and the source of lighting. Therefore, we mainly use road network data as an urban construction factor.

### Carbon emission data

The carbon emission data of China and India in 2015 are from Edgar (https://edgar.jrc.ec.europa.eu/) (accessed on November 20, 2020). We chose the $CO_2$ emission data excluding short-term cycle, which includes all fossil sources of $CO_2$ such as fossil fuel combustion, non-metallic mineral processing (such as cement production), metal (black and non-ferrous metal) production process, urea production, agricultural lime and solvent use. Another short-term sources of $CO_2$ emissions include large-scale biomass burning, tropical grassland burning, forest fires, and land use changes, which were excluded as unstable factors during VIIRS lighting pretreatment. As a result, we do not consider them. Because carbon emissions can reflect the economic structure and energy consumption, many studies have revealed that carbon emissions are one of the important factors affecting NTL [9, 20, 26].

## Methods

In this paper, we regressed the NTL with GDP, population, road network and carbon emissions data from China and India. Then we repeated the experiments at different scales of provinces, cities and counties in China. In addition, we compared the total regression and density regression.

The specific steps were as follows:

1. First, we converted the carbon emission NC format into TIF format with GDAL.

2. Then we defined all TIF data and Shapefile data geographical as coordinates GCS_WGS_1984 and projection coordinates as Asia _ north _ albers _ equal _ area _ conic.

3. Next, we used ARCGIS partition statistics tools to calculate the total NTL, GDP, population and carbon emissions of Indian states and Chinese provinces and counties separately.

4. For the road network, it was necessary to use the ARCGIS identification tool to identify the administrative region to which each road belongs; then we calculated the length of each identified road by using calculation geometry tool, and finally got the total length of the road network of each administrative region according to the administrative region summary tools.

5. The size of the administrative area was also calculated by using calculation geometry tools, and the density data of each parameter were obtained by dividing the total data by the size of the administrative area.

6. Finally, we imported the data processed in ARCGIS into Excel tables or SPSS software for unified sorting and statistical regression analysis. We regressed the NTL with each parameter separately and then with all parameters.

The flow chart is as in Fig 2:

## Results

### Regression in China and India

The relationship between the total amount of NTL and the total amount of various elements of China at a provincial administrative scale is shown in Fig 3.

For the linear regression between each factor and NTL in Fig 3, the P value is less than 0.01, indicating that they have a significant correlation with the NTL. A positive correlation is observed between provincial NTL and all factors in China, and the linear relationship between provincial NTL and GDP is the highest. The multiple regression results show little improvement compared with the linear regression with GDP, and achieve a $R^2$ of 0.883, adjusted $R^2$ of 0.865, and MRE of 28%. The significance of the F test is less than 0.001. It can be seen from Table 2 that GDP is still the main factor explaining NTL at a provincial level in China.

Table 3 shows the Pearson correlation coefficient among various factors. It can be seen that the relationship between GDP and various factors is more compact, while carbon emissions have a lower correlation coefficient with other factors. The significance of all parameters is less than 0.01.

The density sometimes has different meanings with total amount, and it is also used in many situations. Therefore, we divide all economic factors and NTL by the area of the administrative region and then do regression and correlation analyses. This result is shown in Fig 4.

We can see that there are some points with high values of various parameters in regression. They are megacities such as Shanghai, Tianjin, and Beijing [44, 45]. These points usually have

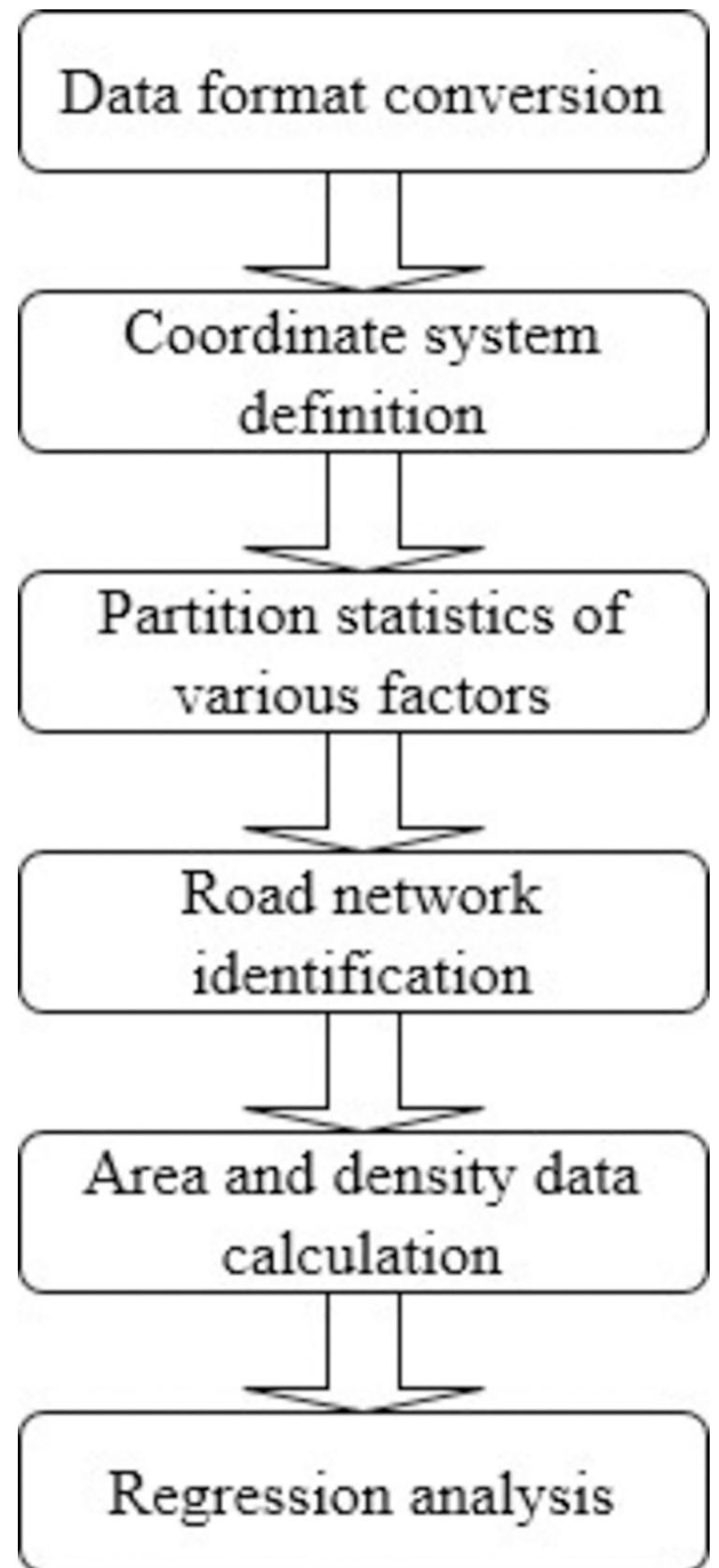

**Fig 2. Flow chart of experimental steps.**

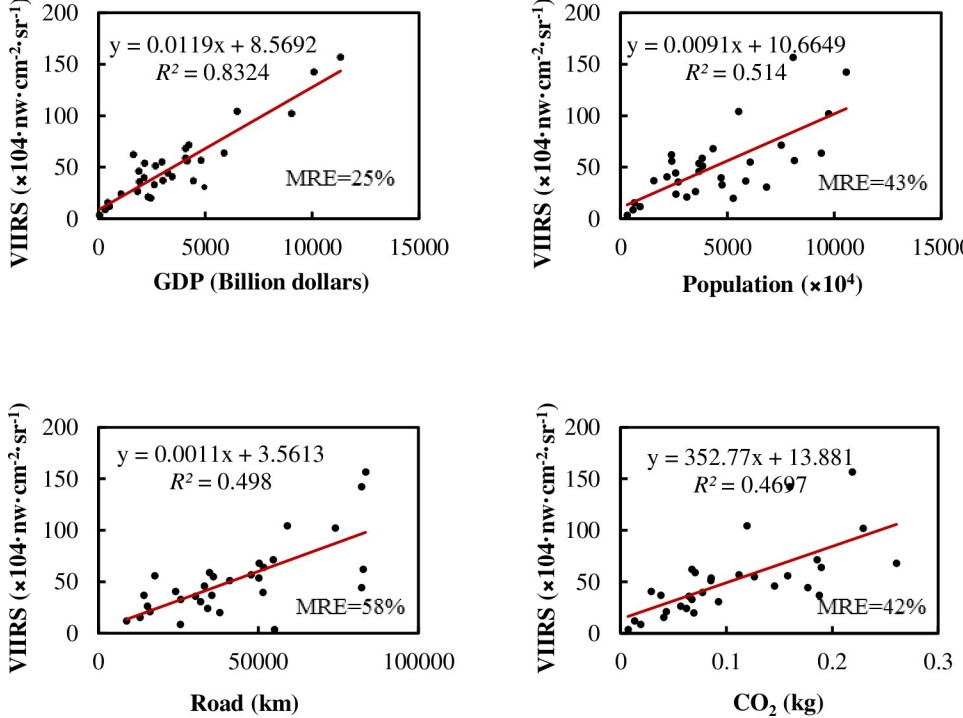

**Fig 3. Relationship between total NTL and various elements at the provincial administrative scale in China.**

a better linear relationship between NTL and various factors, and have a greater impact on the regression after being normalized with the administrative area. If these megacities are excluded, the regression results are shown in Fig 5. The $R^2$ of linear regression between the NTL density and the density of each factor is still improved compared with the total regression results, even after excluding the three megacities, but the MRE also becomes larger. It seems that the developed small-scale provinces in eastern China will bring a better linear trend in the regression, normalized with the administrative area, even after excluding three megacities. However, these provinces with large areas and low lights still make the MRE bigger. So, if we use the model to calculate the NTL in the central and western provinces, there will still be a certain relative error.

The multiple regression of all regions achieves $R^2$ of 0.998, adjusted $R^2$ of 0.998, and MRE of 37%. The significance of the F test is less than 0.001. Detailed regression results of all variables with the NTL are shown in Table 4. After being normalized with the administrative area, the importance of some factors, such as carbon emissions, improves compared with the total multiple regression. As for Pearson's correlation coefficient, it is relatively high. If we exclude the three megacities, the multiple regression achieves $R^2$ of 0.978, adjusted $R^2$ of 0.975, and MRE of 29%, and the significance of the F test is less than 0.001. See Table 5 for the weight

**Table 2. Multiple regression coefficient at provincial level in China.**

| variable | Standard coefficient | t | P value |
|:---:|:---:|:---:|:---:|
| GDP | 0.873 | 6.673 | 0.000 |
| POP | -0.169 | -1.364 | 0.184 |
| ROAD | 0.257 | 2.967 | 0.006 |
| $CO_2$ | 0.039 | 0.381 | 0.706 |

**Table 3. Pearson's correlation coefficient between the total NTL and each factor at provincial level in China.**

|  | NTL | GDP | POP | ROAD | $CO_2$ |
|---|---|---|---|---|---|
| NTL | 1 | - | - | - | - |
| GDP | 0.912 | 1 | - | - | - |
| POP | 0.717 | 0.827 | 1 | - | - |
| ROAD | 0.706 | 0.593 | 0.535 | 1 | - |
| $CO_2$ | 0.685 | 0.707 | 0.687 | 0.568 | 1 |

coefficients and Pearson's correlation coefficients. Although the $R^2$ decreases slightly after we exclude the three megacities, the MRE is good, which makes it more valuable to predict factors. At this time, the road network occupies a more important position.

For the linear regression between each factor and NTL in India (Fig 6), the P value is still less than 0.01. Similar to the regression of provincial totals in China, there is a positive correlation between NTL and several basic elements. However, there are some differences in terms of specific details. Compared with China, India's NTL has a lower $R^2$ with GDP, but a higher $R^2$ with population, road network, and carbon emissions. The MRE is generally on the high side. In addition, after multiple regression in India, the accuracy is significantly improved, $R^2$ is 0.938, adjusted $R^2$ is 0.929, MRE is 49%, and F test is less than 0.001. The multiple regression results show great improvement compared with the respective regression, which is different from China. See Table 6 for the weight coefficients and Pearson's correlation coefficients.

Table 6 is obviously different from Tables 2 and 3 in China. In China's multiple regression, GDP still occupies a dominant position, while the other three factors are almost of little importance and have no obvious effect on improving the precision. However, after India's multiple

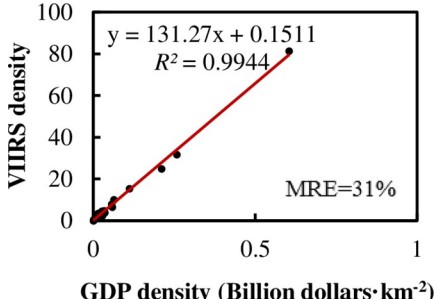
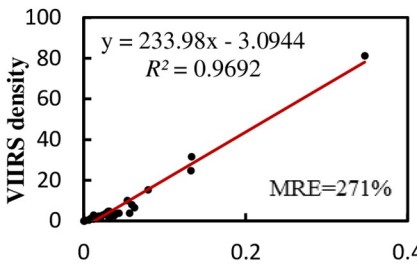

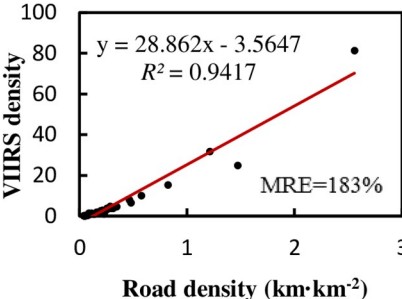
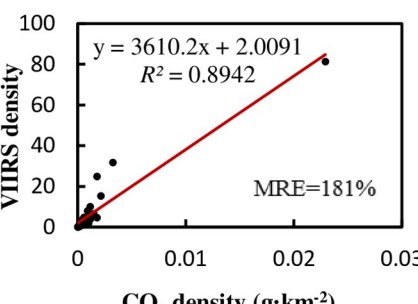

**Fig 4. Relationship between NTL density and the density of various elements in provincial administrative regions of China.**

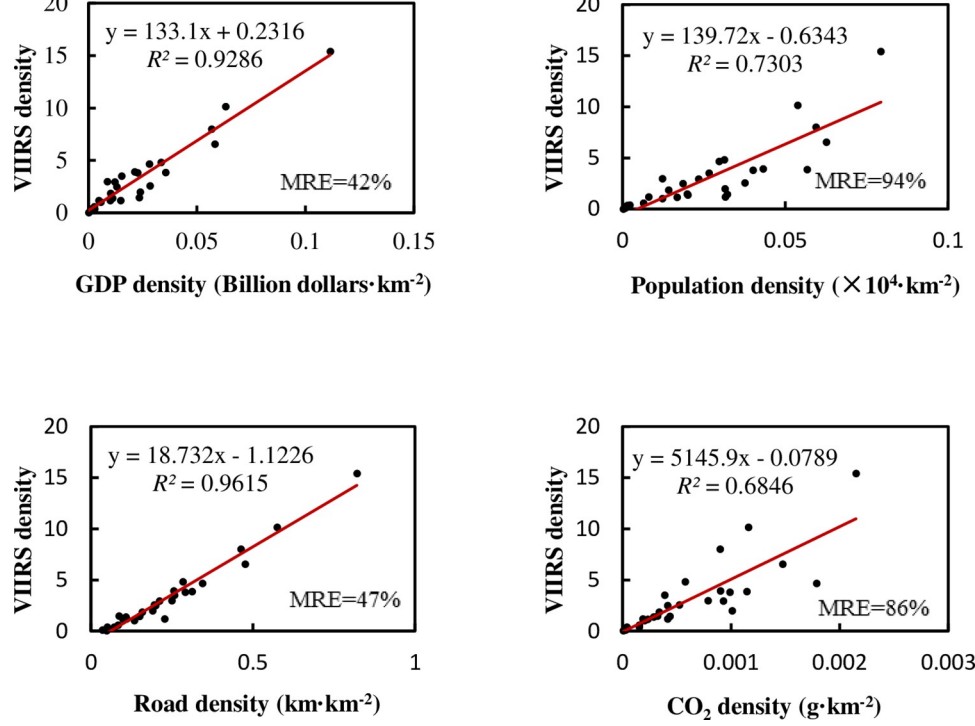

**Fig 5. Relationship between NTL density and the density of various elements in provincial administrative regions of China after excluding three megacities.**

regression, the importance of GDP has greatly decreased, while population, road network and carbon emissions have become important factors, and the final accuracy has obviously improved. The Pearson correlation between these factors is a little different from that in China: the relationship between NTL and population is slightly stronger, but the relationships between NTL and GDP, and GDP and population are slightly weaker. In addition, the relationship between carbon emissions, road network and other factors is obviously stronger.

If divided by the size of the administrative area, using density to regress, as shown in Fig 7:

**Table 4. Multiple regression coefficient and Pearson's correlation coefficient of NTL density at provincial level in China.**

| (a) Multiple regression coefficient. | | | | |
|---|---|---|---|---|
| **Predict** | **Standard coefficient** | **t** | **P value** | |
| GDP | 0.764 | 10.706 | 0.000 | |
| POP | -0.178 | -2.584 | 0.016 | |
| ROAD | 0.210 | 3.520 | 0.002 | |
| $CO_2$ | 0.219 | 6.400 | 0.000 | |
| (b) Pearson's correlation coefficient. | | | | |
| | **NTL** | **GDP** | **POP** | **ROAD** | **$CO_2$** |
| NTL | 1 | - | - | - | - |
| GDP | 0.997 | 1 | - | - | - |
| POP | 0.984 | 0.987 | 1 | - | - |
| ROAD | 0.970 | 0.975 | 0.975 | 1 | - |
| $CO_2$ | 0.946 | 0.930 | 0.928 | 0.861 | 1 |

**Table 5. Multiple regression coefficient and Pearson's correlation coefficient of NTL density at provincial level in China after excluding three megacities.**

| (a) Multiple regression coefficient. | | | | |
|---|---|---|---|---|
| **Predict** | **Standard coefficient** | **t** | **P value** | |
| GDP | 0.437 | 3.547 | 0.002 | |
| POP | -0.218 | -2.740 | 0.012 | |
| ROAD | 0.810 | 6.432 | 0.000 | |
| $CO_2$ | -0.060 | -0.909 | 0.373 | |
| (b) Pearson's correlation coefficient | | | | |
| | **NTL** | **GDP** | **POP** | **ROAD** | **$CO_2$** |
| NTL | 1 | - | - | - | - |
| GDP | 0.964 | 1 | - | - | - |
| POP | 0.855 | 0.914 | 1 | - | - |
| ROAD | 0.981 | 0.958 | 0.894 | 1 | - |
| $CO_2$ | 0.827 | 0.825 | 0.828 | 0.875 | 1 |

Similar to China, when we use density data to regress, various regression $R^2$ of India is obviously improved. The regression is also affected by outliers. After excluding two megacities, the regression result is as shown in the Fig 8:

It can be seen that the regression accuracy between NTL and various factors is poor after excluding two megacities. Not only does $R^2$ becomes lower, but MRE also increases. Initially, the linear relationship between the total NTL and GDP in India was not as good as in China, because India's regional development is more uncoordinated. After removing some megacities, the $R^2$ of India's density regression drops rapidly.

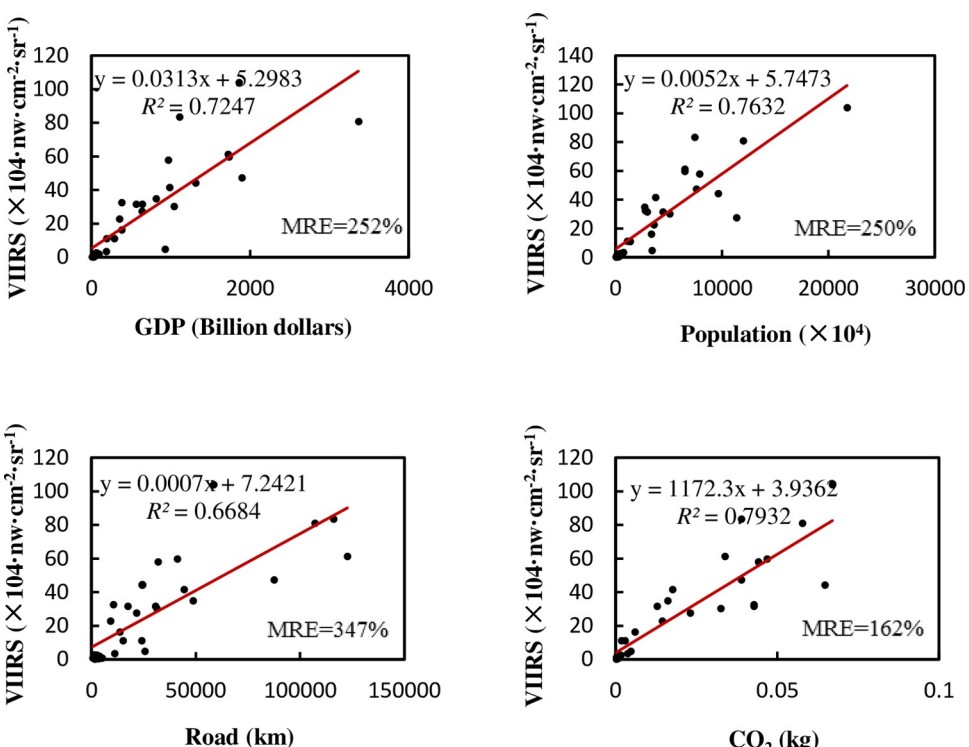

**Fig 6. Relationship between total NTL and various elements in Indian state-level administrative regions.**

**Table 6. Multiple regression coefficient table and Pearson correlation coefficient table of total NTL amount in Indian state.**

| (a) Multiple regression coefficient | | | | |
|---|---|---|---|---|
| **Predict** | **Standard coefficient** | **t** | **Significant** | |
| **GDP** | -0.161 | -1.344 | 0.190 | |
| **POP** | 0.346 | 3.643 | 0.001 | |
| **ROAD** | 0.431 | 4.225 | 0.000 | |
| **$CO_2$** | 0.475 | 5.433 | 0.000 | |
| (b) Pearson's correlation coefficient | | | | |
| | **LIGHT** | **GDP** | **POP** | **ROAD** | **$CO_2$** |
| **LIGHT** | 1 | - | - | - | - |
| **GDP** | 0.851 | 1 | - | - | - |
| **POP** | 0.874 | 0.776 | 1 | - | - |
| **ROAD** | 0.818 | 0.831 | 0.602 | 1 | - |
| **$CO_2$** | 0.891 | 0.808 | 0.850 | 0.621 | 1 |

The accuracy of the multiple regression of all regions achieves $R^2$ of 0.994, adjusted $R^2$ of 0.993, and MRE of 55%, and the significance of the F test is less than 0.001. We can still understand the relationship among the factors by multiple regression (Table 7). After excluding two megacities, the multiple regression achieves $R^2$ of 0.791, adjusted $R^2$ of 0.757, and MRE of 62%, and the significance of the F test is less than 0.001. Although the regression accuracy is still not very good, it is obviously improved compared with the respective regression. From Table 8's weight coefficients and Pearson's correlation coefficients, we can see that the road network plays an important role, just as it did in the normalized multiple regression in China.

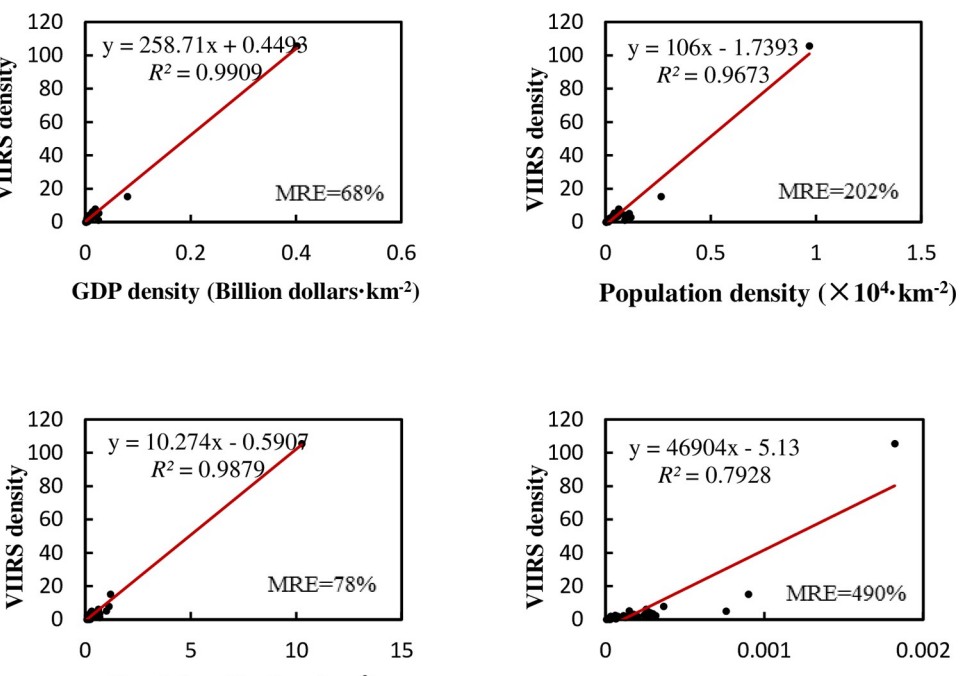

**Fig 7. Regression relationship between NTL density and density of various elements in state-level administrative regions of India.**

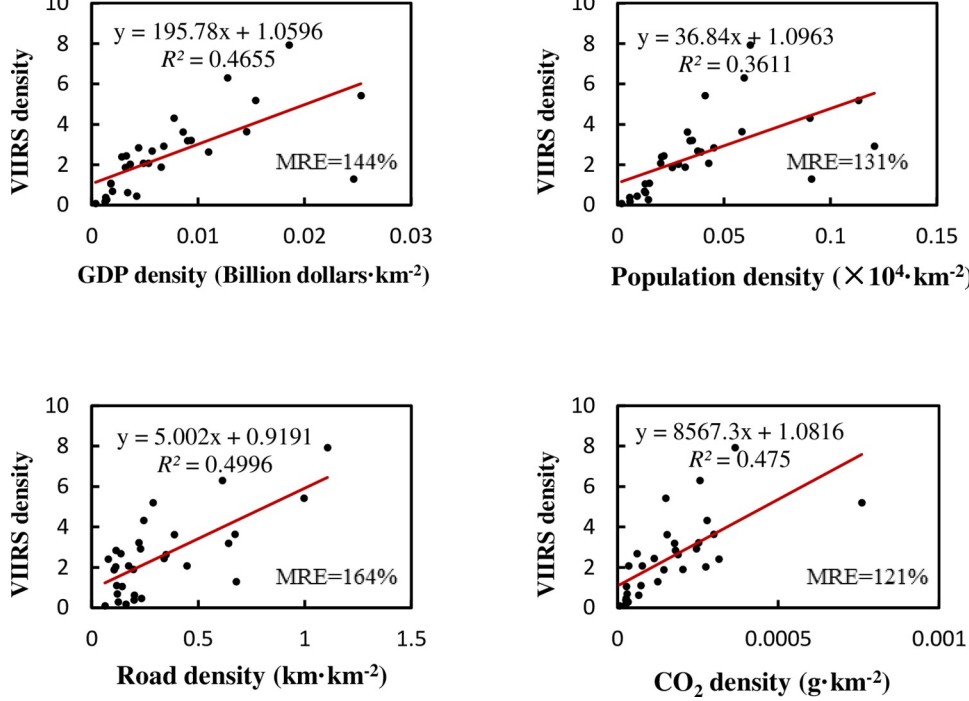

**Fig 8. Regression relationship between NTL density and density of various elements in state-level administrative regions of India after excluding two megacities.**

Compared with China, the importance of GDP in explaining NTL has declined in India, while the carbon emissions density has a higher influence.

Whether in the regression of total amount or the regression normalized with region area, none of the factors can uniformly measure the NTL of the two countries. The relationship between NTL and each factor is obviously different in the two countries. For example, GDP has different regression coefficients with NTL in China and India. The difference of the coefficient of GDP and NTL is probably affected by the ratios of several other important factors. Therefore, we chose GDP and road network, which are very influential in both China and India. Then, all the administrative regions of the two countries were used to regress with total

**Table 7. Multiple regression coefficient table and Pearson correlation coefficient table of NTL density in Indian state.**

| (a) Multiple regression coefficient | | | | |
|---|---|---|---|---|
| **Predict** | **Standard coefficient** | **t** | **Significant** | |
| **GDP** | 0.329 | 1.612 | 0.119 | |
| **POP** | 0.099 | 0.741 | 0.465 | |
| **ROAD** | 0.532 | 4.028 | 0.000 | |
| **$CO_2$** | 0.045 | 1.025 | 0.315 | |
| (b) Pearson's correlation coefficient | | | | |
| | **LIGHT** | **GDP** | **POP** | **ROAD** | **$CO_2$** |
| **LIGHT** | 1 | - | - | - | - |
| **GDP** | 0.995 | 1 | - | - | - |
| **POP** | 0.984 | 0.989 | 1 | - | - |
| **ROAD** | 0.994 | 0.992 | 0.972 | 1 | - |
| **$CO_2$** | 0.890 | 0.894 | 0.929 | 0.862 | 1 |

**Table 8. Multiple regression coefficient table and Pearson correlation coefficient table of NTL density in Indian state after excluding two megacities.**

| (a) Multiple regression coefficient | | | | | |
|---|---|---|---|---|---|
| **Predict** | **Standard coefficient** | **t** | **Significant** | | |
| **GDP** | -0.349 | -1.556 | 0.132 | | |
| **POP** | 0.160 | 1.060 | 0.299 | | |
| **ROAD** | 0.806 | 4.267 | 0.000 | | |
| **CO$_2$** | 0.528 | 4.084 | 0.000 | | |
| (b) Pearson's correlation coefficient | | | | | |
| | **LIGHT** | **GDP** | **POP** | **ROAD** | **CO$_2$** |
| **LIGHT** | 1 | - | - | - | - |
| **GDP** | 0.682 | 1 | - | - | - |
| **POP** | 0.601 | 0.616 | 1 | - | - |
| **ROAD** | 0.707 | 0.848 | 0.354 | 1 | - |
| **CO$_2$** | 0.689 | 0.474 | 0.702 | 0.267 | 1 |

NTL. The regression precision achieves $R^2$ of 0.824, adjusted $R^2$ of 0.818, and MRE of 68%, and the significance of F test is less than 0.001. The weight coefficient is shown in Table 9. It is far better than just regression with GDP in these two countries, which only achieve a $R^2$ of 0.598.

It can be seen that, although China and India have different coefficients for various regressions, they can be uniformly measured in multiple regression from the perspective of $R^2$.

## Comparison of province and county scale regressions in China

The relationship between the total NTL with the total GDP at China's province, city, county, and town scales reveals that the relationship at the city level was similar to that at the provincial level, with $R^2$ reaching 0.84. However, $R^2$ drops rapidly at the county and town levels, reaching 0.63 and 0.59, respectively. Because there is no obvious difference between the city level and the provincial level, and the accuracy of some data at the township level is hard to guarantee, we selected the county level and the province level for comparison and further analysis.

The relationship between total NTL and various elements at the county level in China is shown in Fig 9.

It can be seen from Fig 9 that NTL is still positively correlated with GDP, population, and road network. The relationship between NTL and carbon emissions does not have a linear correlation. In addition, the regression coefficient of various parameters changed slightly compared with the provincial level in China, especially the correlation coefficient with population. The multiple regression result is shown in Table 10.

The weight of GDP in explaining NTL is greatly reduced and other factors besides carbon emissions play a role in the multiple regression, which is very different from the province level multiple regression. The multiple regression achieves $R^2$ of 0.733, adjusted $R^2$ of 0.733, and MRE of 267%. Although the accuracy is still not good, it is improved compared with the univariate regression, and the improvement is more than that of the provincial multiple regression. In addition, Pearson's correlation coefficients are also worse than provincial multiple regressions.

**Table 9. Multiple regression coefficient of total NTL in China and India.**

| **Predict** | **Standard coefficient** | **t** | **P value** |
|---|---|---|---|
| **GDP** | 0.566 | 9.423 | 0.000 |
| **ROAD** | 0.519 | 8.646 | 0.000 |

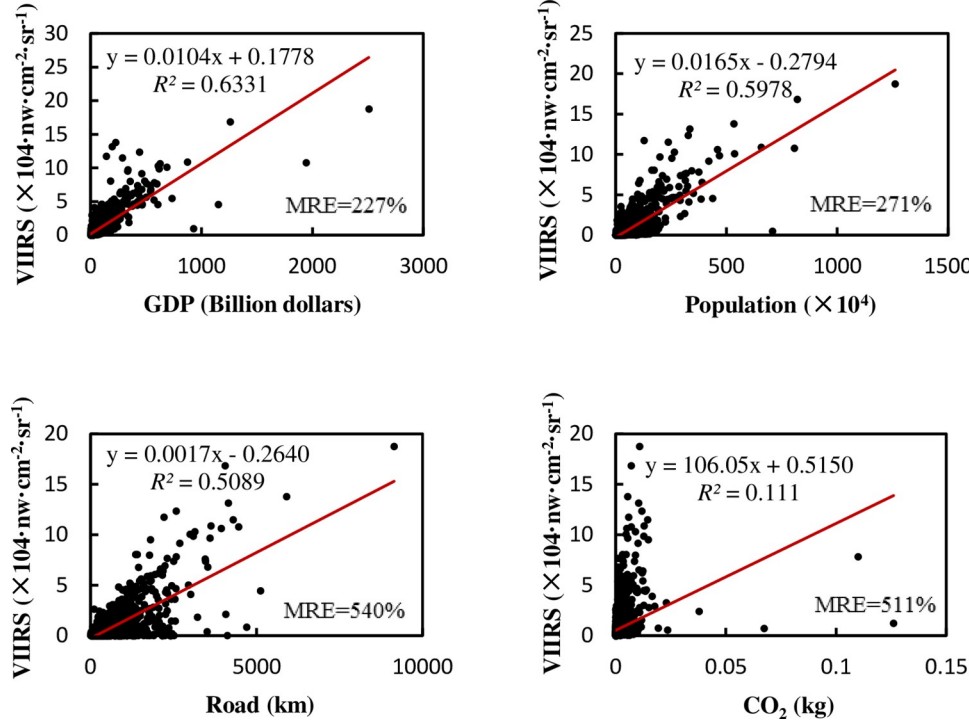

**Fig 9. Relationship between total NTL and total elements in county-level administrative regions of China.**

After dividing by the area of the administrative region, the density regression results are as shown in Fig 10.

It can be seen that the $R^2$ improvement when using density data disappeared at a county level. Only the $R^2$ of population and road network density showed slight improvement. It should be noted that, although the linear trend has no changed much, the MRE is too large. We find that many points are near the origin of the coordinates, which will cause huge MRE. We took the double logarithm of both horizontal and vertical coordinates, and the regression results are as shown in the Fig 11:

**Table 10. Multiple regression coefficient and Pearson's correlation coefficient of total NTL quantity at county level in China.**

| (a) Multiple regression coefficient | | | | |
|---|---|---|---|---|
| **Predict** | **Standard coefficient** | **t** | **Significant** | |
| **GDP** | 0.348 | 17.087 | 0.000 | |
| **POP** | 0.276 | 14.034 | 0.000 | |
| **ROAD** | 0.306 | 21.909 | 0.000 | |
| **$CO_2$** | 0.074 | 6.617 | 0.000 | |
| (b) Pearson's correlation coefficient | | | | |
| | **LIGHT** | **GDP** | **POP** | **ROAD** | **$CO_2$** |
| **LIGHT** | 1 | - | - | - | - |
| **GDP** | 0.796 | 1 | - | - | - |
| **POP** | 0.773 | 0.837 | 1 | - | - |
| **ROAD** | 0.713 | 0.640 | 0.600 | 1 | - |
| **$CO_2$** | 0.333 | 0.281 | 0.300 | 0.258 | 1 |

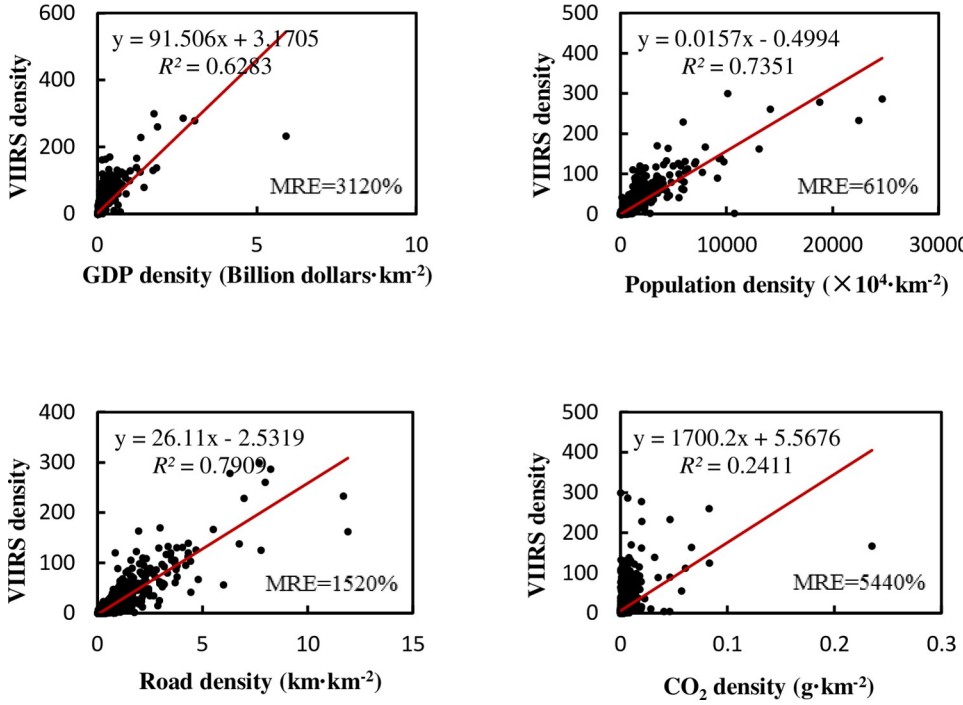

**Fig 10. Relationship between NTL density and density of various elements in county-level administrative regions of China.**

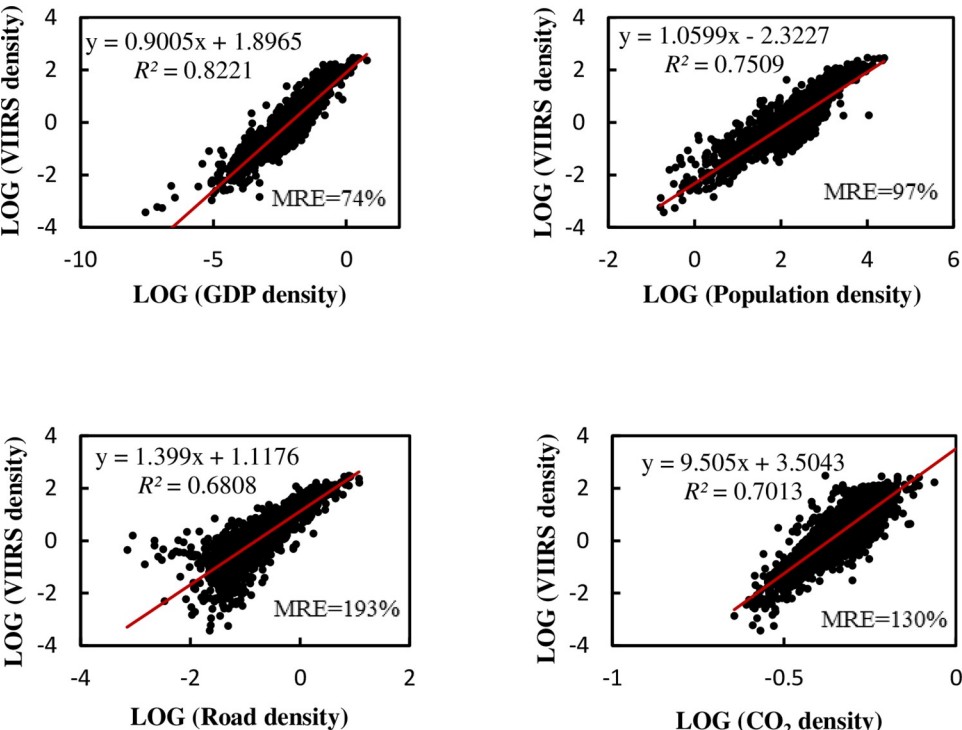

**Fig 11. Relationship between double logarithmic NTL density and density of various elements in county-level administrative regions of China.**

**Table 11. Multiple regression coefficient and Pearson's correlation coefficient of NTL density at county level in China.**

| (a) Multiple regression coefficient | | | | |
|---|---|---|---|---|
| **Predict** | **Standard coefficient** | **t** | **Significant** | |
| **GDP** | 0.072 | 4.133 | 0.000 | |
| **POP** | 0.303 | 14.891 | 0.000 | |
| **ROAD** | 0.547 | 33.659 | 0.000 | |
| **CO$_2$** | 0.062 | 6.472 | 0.000 | |
| (b) Pearson's correlation coefficient | | | | |
| | **LIGHT** | **GDP** | **POP** | **ROAD** | **CO$_2$** |
| **LIGHT** | 1 | - | - | - | - |
| **GDP** | 0.793 | 1 | - | - | - |
| **POP** | 0.857 | 0.873 | 1 | - | - |
| **ROAD** | 0.889 | 0.787 | 0.847 | 1 | - |
| **CO$_2$** | 0.491 | 0.421 | 0.461 | 0.474 | 1 |

Obviously, the regression accuracy has greatly improved. Not only does $R^2$ become much higher, but MRE also returns to normal.

The linear multiple regression $R^2$ is 0.833, the adjusted $R^2$ is 0.833, and the MRE is 1490%, which is better than the univariate regression. However, the MRE is still too high, so it has no meaning and is not suitable for forecasting. The multiple regression results are shown in Table 11. After taking the double logarithm, the multiple regression achieves $R^2$ of 0.876, adjusted $R^2$ of 0.876, and MRE of 53%. The accuracy is greatly improved, and the regression accuracy with carbon emissions becomes good. The regression results are as shown in Table 12.

It can be seen that the population and road network play a major role in a linear multiple regression, while the importance of GDP has dropped greatly compared to Table 10. In the double logarithmic multiple regression, GDP returns to be the major factor.

## Discussion

The total data regression accuracy of county-level NTL with GDP was significantly lower than at the provincial level in China, while the $R^2$ for population and road network were slightly improved. We think it was a matter of integration: in China, a province is a complete entity,

**Table 12. Multiple regression coefficient and Pearson's correlation coefficient of double logarithmic NTL density at county level in China.**

| (a) Multiple regression coefficient | | | | |
|---|---|---|---|---|
| **Predict** | **Standard coefficient** | **t** | **Significant** | |
| **GDP** | 0.528 | 23.047 | 0.000 | |
| **POP** | 0.008 | 0.367 | 0.714 | |
| **ROAD** | 0.280 | 24.449 | 0.000 | |
| **CO$_2$** | 0.189 | 13.721 | 0.000 | |
| (b) Pearson's correlation coefficient | | | | |
| | **LIGHT** | **GDP** | **POP** | **ROAD** | **CO$_2$** |
| **LIGHT** | 1 | - | - | - | - |
| **GDP** | 0.907 | 1 | - | - | - |
| **POP** | 0.867 | 0.941 | 1 | - | - |
| **ROAD** | 0.825 | 0.761 | 0.729 | 1 | - |
| **CO$_2$** | 0.837 | 0.831 | 0.829 | 0.721 | 1 |

which has balanced development in various fields. There is also a high Pearson correlation among the parameters. GDP is a comprehensive economic indicator, more representative for NTL when regions have better integration. If the regional development structure is unbalanced, the regression accuracy with GDP will be reduced. The population and road network are not comprehensive indicators but direct indicators from a certain angle, so the regression $R^2$ will not be lower.

China's counties are relatively scattered administrative units, which does not have a complete integration, and get a lower Pearson correlation coefficient among the parameters. They often have obvious differences in characteristics (for example, some counties mainly develop their tourism and catering industries, while others develop commerce or industry). So, the regression accuracy of NTL and GDP becomes worse. At this time, a multiple regression can usually improve the accuracy. It achieves an $R^2$ of 0.733 when using the total NTL data, which is higher than that of a separate regression with GDP. This is also the case in the comparison of provincial experiments between India and China, which may be a similar reason. After all, the imbalance between Indian States is more serious than that of China [39–41].

However, compared with India's multiple regression at state level, where $R^2$ directly reaches 0.938, it is still far worse. In fact, China's provincial multiple regression $R^2$ value is 0.883, which is not as good as India's either. There are many reasons for this, but we can distinguish two as the most important. The first is the model. We used to try four models in experiments: linear, exponential, logarithmic and double logarithmic. Most of the time, the linear model is the most suitable, so we mainly keep the result images of linear regression in the paper. However, some special circumstances may make logarithmic and double logarithmic models more suitable. Lighting in developed cities tends to underestimate GDP and population, while in backward areas and rural areas it tends to overestimate GDP and population. Therefore, some researchers suggest using a double logarithmic function, polynomial function, or sigmoid function to regress [31, 32, 46]. In this study, we can see that there are many points close to the origin of coordinates at the county level in China, They cannot adapt to the linear relationship well. The NTL is often underestimated, and these points are discrete to cause large MRE. After normalization with the administrative area, the MRE become very large. This is because there is an obvious development gap between eastern and western China, and while many counties' NTL is low, the area is not small. After taking the double logarithm, not only does $R^2$ become 0.876, but MRE also returns to normal. If we choose a model that adapts to the difference of NTL distribution in China, such as graded regression, and consider the spatial structure of city and light area [47, 48], the effect may be better. In fact, the light intensity sometimes cannot reflect the population well, but the spatial concentration of the population can be reflected in the light area. In addition, the carbon emissions must be double logarithmic before the regression can achieve meaningful results. The second reason is the factors. When the accuracy of a regression with GDP is not high enough, we use a multiple regression to compensate. The four factors we choose are often used in the NTL analysis [19–24], but they may not reflect all aspects of Chinese NTL. The multiple regression $R^2$ in China always cannot reach above 0.9. Some factors often fail to play a good role in China's multiple regression, such as carbon emissions. The reason for choosing carbon emissions as a parameter is that we think it can represent the GDP structure and energy consumption. Cities with a similar GDP can be of different types, such as industrial cities and commercial cities. They will have different carbon emissions and NTL. In addition, carbon emissions represent energy consumption, which is positively correlated with NTL. Generally, the relationship between carbon emissions and NTL is complex and related to national conditions. In fact, carbon emissions have a good relationship with NTL and play a key role in the multiple regression in India. After removing carbon emissions, India's multiple regression can only reached an $R^2$ value of 0.897. However, the carbon

emissions are not good predictors in China. One of the reasons for this may be that China's transmission facilities are more developed than India's [34, 35], and areas with high carbon emissions are often not areas with high NTL. Many large-scale thermal power plants are located in Inner Mongolia, Shanxi, and other areas where there are raw materials. The power they generate is exported to all parts of the country [49]. At the same time, China's nuclear power, hydropower, and wind power are also developed [50]. And from the experimental data, we can see that China's carbon emissions are so high that NTL can't explain its main components. The logarithmic regression effect of carbon emissions and NTL is obviously better than linear regression, which indicates its saturation phenomenon. Combined with many factors, the heavy industrial areas with high carbon emissions cannot be well matched with the high NTL areas. China's county level is also affected by this. Moreover, as piecemeal administrative units, China's county-level GDP representativeness becomes worse. The lack of elements has a bigger impact on the accuracy. If carbon emissions are replaced by more direct elements like electric power consumption [51], the effect of the multivariate regression may be greatly improved. However, unfortunately, China's county-level electric power consumption data are not easy to obtain.

We can see that the economic development level of the two countries cannot be uniformly measured by NTL. The regression has different coefficients in the two countries. We tried to find the reasons for the differences in two important factors, GDP and road network. The multiple regression achieves good results. This idea can be popularized, but the parameter may be different in different countries.

At the provincial level in both China and India, the effect of using density is much better than using the total amount when regressed with GDP. There should not be such a big change in our estimates. Using density data divided by the area is only a stretch. $R^2$ should not change much if every point is stretched almost equally. However, experiments now show that stretching is obviously unbalanced, which will significantly improve $R^2$ in China and India. After careful exploration, we find an important reason is that the NTL and urban construction have obvious spatial clustering characteristics. Some highly developed regions are usually small in area, but they are classified as a separate administrative region, such as Shanghai, Tianjin, Beijing, and Jiangsu in China, Chandigarh and Puducherry in India, etc [44, 45, 52]. From the experimental data, it can be seen that these megacities, for which the GDP density is almost above 10 times that the average of other regions, show a more linear relationship between NTL and GDP or other factors, because these regions are more balanced and unified in their development, and they have a better Pearson correlation of each parameter. Although these regions are developed, they are small in area and have little influence on the total data's regression. However, after dividing by the area, they have a great influence and make $R^2$ higher.

It can be seen that, although a high $R^2$ value is reflected in the density regression, this is mainly due to the good linearity between the NTL and GDP in some megacities, and there is often a larger MRE as well. Therefore, when we use density data, it is incorrect to directly estimate the GDP of many ordinary regions by NTL even with high $R^2$ on the surface, especially in a country with unbalanced development. In this case, a multiple regression can often lead to better accuracy. In fact, after excluding several megacities, India's state-level multiple regression, after normalization with the administrative area, can still maintain an $R^2$ value of 0.791, which is obviously higher than the separate regression with GDP. At the same time, multiple regression using GDP and road network density data from all regions in China and India can also maintain an $R^2$ of value 0.830.

At a county level in China, the regression accuracy between NTL and GDP has not improved as much as at the provincial level when using density data. This is in line with the above reasons. Because After all districts are further divided to county level, the influence of

megacities has also been lost. At this time, these highly developed small counties have lost the influence of megacities such as Beijing and Shanghai in a large number of counties when using density data to regress, and the regression accuracy of NTL will not improve with GDP.

We have got some valuable results from experiments, but there are also some defects. Our research is not sufficient in time and space. It would be very valuable if further consider the changes of some factors in a longer time range and then conduct these experiments to see the change of NTL [53].

## Conclusions

In this paper, we compared the relationships between NTL and various elements in China and India, and repeated the experiments between the provinces and counties of China. Then we selected the total data and density data for regression. From the experiments, we know that the GDP, as a comprehensive index, has a good linear correlation with NTL when we do the regression in administrative regions with strong integrity. If administrative units are fragmented or unbalanced in development, other factors such as the population and road network will play an important role in multiple regression to improve the $R^2$. The NTL differences between different countries can also be explained by these additional important factors. Regression after normalization with the administrative area usually has a better $R^2$ value than total amount regression, especially at a provincial (state) level. The main reason is that there is a better linearity between NTL and various elements in highly developed regions, and these megacities have a more significant impact in regressions after being divided by the area. These basic conclusions can help us to use NTL more rationally for the economic analysis of different cities in a wide range, and adjust NTL in some cases to improve the accuracy of the analysis. However, there are many shortcomings in this paper, which deserve further discussion. First of all, when selecting parameters, we think that many of the lights received by satellites may come from light sources outside the streets. So, we choose the road network. However, this can also be compared with the building area and height so as to know more about the proportion of street lights and indoor lights. In addition, carbon emissions, as an indicator of GDP structure and energy consumption, obviously do not play a positive role in China. The next step is to find a better indicator, like electricity consumption or generation, to replace carbon emissions. Secondly, although this paper compares various factors, it does not pay more attention to the model itself. In fact, some problems may not need multiple factors, but a more correct model can bring about similar or even better results. Studies have shown that the relationship between NTL and GDP is not always linear. After the model is improved, what role each factor will play in different countries or different scales is a question worthy of further discussion.

## Supporting information

**S1 Data.**
(RAR)

## Acknowledgments

Thanks to Professor QIN Qiming and REN Huazhong for their guidance.

## Author Contributions

**Data curation:** Guhuai Han, Shoujie Zhu.

**Funding acquisition:** Tao Zhou, Yuanheng Sun.

**Methodology:** Guhuai Han.

**Project administration:** Yuanheng Sun.

**Resources:** Shoujie Zhu.

**Software:** Shoujie Zhu.

**Supervision:** Guhuai Han, Yuanheng Sun.

**Validation:** Guhuai Han.

**Visualization:** Guhuai Han.

**Writing – original draft:** Guhuai Han.

**Writing – review & editing:** Guhuai Han, Tao Zhou, Yuanheng Sun.

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
