## [Decision Letter · Decision Letter 0]

12 Jul 2021

PONE-D-21-17224

The relationship between night-time light and socioeconomic factors in China and India

PLOS ONE

Dear Dr. Sun,

Thank you for submitting your manuscript to PLOS ONE. After careful consideration, we feel that it has merit but does not fully meet PLOS ONE’s publication criteria as it currently stands. Therefore, we invite you to submit a revised version of the manuscript that addresses the points raised during the review process.

After taking a look at the reviewer' comments and your manuscript, I recommend a major revision. You should provide more results that show the rationality of selecting the economic indicators (e.g., road network data).

We look forward to receiving your revised manuscript.

Kind regards,

Wenhao Yu, Ph.D.

Academic Editor

PLOS ONE

Additional Editor Comments:

After taking a look at the reviewer' comments and your manuscript, I recommend a major revision. You should provide more results that show the rationality of selecting the economic indicators (e.g., road network data).

Journal Requirements:

3. We note that Figure 1 in your submission contain map images which may be copyrighted. All PLOS content is published under the Creative Commons Attribution License (CC BY 4.0), which means that the manuscript, images, and Supporting Information files will be freely available online, and any third party is permitted to access, download, copy, distribute, and use these materials in any way, even commercially, with proper attribution. For these reasons, we cannot publish previously copyrighted maps or satellite images created using proprietary data, such as Google software (Google Maps, Street View, and Earth). For more information, see our copyright guidelines: http://journals.plos.org/plosone/s/licenses-and-copyright.

4. We note that you currently have two tables named "table 7". Please correct it accordingly.

Reviewers' comments:

Reviewer's Responses to Questions

**Comments to the Author**

1. Is the manuscript technically sound, and do the data support the conclusions?

Reviewer #1: Yes

2. Has the statistical analysis been performed appropriately and rigorously? 

Reviewer #1: Yes

3. Have the authors made all data underlying the findings in their manuscript fully available?

Reviewer #1: Yes

4. Is the manuscript presented in an intelligible fashion and written in standard English?

Reviewer #1: Yes

5. Review Comments to the Author

Reviewer #1: This paper uses remote sensing data - nighttime light (NTL) as well as a rang of economic parameters for China and India to reveal the intrinsic link between the two types of data at different scales. The subject is interesting, as are the results, although there a number of issues and limitations that should be addressed before publication. These comments are presented below.

Main comments:

1.I am concerned with the choice of spatial units and the analysis carried out by these choices. The authors seem to have carried out three scales, namely the city level, the province level and the national level. However, the administrative level of a city or province is sometimes not indicative of its status in terms of economic level, e.g. the capital city of China (Beijing) may have a higher level of economic development as well as NTL than some provinces. I would recommend including a bit more literature and taking into account the economic ranking of the city or province. See, for example Guo, Y., Yu, W., Chen, Z., & Zou, R. (2020).

2.Did the authors select the cities or provinces that participated in the experiment? Because some cities or provinces have very low light values, they do not effectively represent the state of their economic development. I recommend that the authors give at least some summary description of the cities and provinces that participated in the experiment.

3.What is the author's rationale for selecting these economic indicators (GDP, Population, Road network data, Carbon emission data)? In particular, why is road network data used as a type of economic indicator, when it seems that road network data is hardly representative of the economic development of a region in general, e.g. Switzerland does not have an extremely well developed road network (metro, etc.) in most cantons, but its cities still have a high economic level.

4.Have the authors considered the possibility that the NTL data and several of the data in the text are other regression relationships, for example (exponential relation)? Because the level of socioeconomic development of the region does not always show linear growth, at different times in the development of the region, especially in the later stages of development, the economic level of the region gradually slows down and shows saturation levels. Some experiments on this aspect should be added to this paper as this may affect this paper’s conclusion.

Minor comments:

1.Page 5. A table with the type of data, a short description and the source of the data is added to the data section to make the article more readable.

2.Page 17. Regional differences between China and India (e.g. infrastructure development, population, etc.) should be added to the discussion section, and it is possible that these also contribute to the different return relationships.

6. PLOS authors have the option to publish the peer review history of their article (what does this mean?). If published, this will include your full peer review and any attached files.

Reviewer #1: No

---

## [Author Response · Author response to Decision Letter 0]

30 Sep 2021

Dear editor

 Thank you for your reminder，We have uploaded the data as a Supporting Information file.

Sincerely yours,

Han Guhuai

---

## [Decision Letter · Decision Letter 1]

29 Nov 2021

PONE-D-21-17224R1The relationship between night-time light and socioeconomic factors in China and IndiaPLOS ONE

Dear Dr. Sun,

Thank you for submitting your manuscript to PLOS ONE. After careful consideration, we feel that it has merit but does not fully meet PLOS ONE’s publication criteria as it currently stands. Therefore, we invite you to submit a revised version of the manuscript that addresses the points raised during the review process.

We look forward to receiving your revised manuscript.

Kind regards,

Jun Yang

Academic Editor

PLOS ONE

Journal Requirements:

Additional Editor Comments:

The authors explored the relationship between night-time light and socioeconomic factors. The research methodologies are reasonable, and the findings are justifiable. However, there are still a few aspects that should be improved to make the paper publishable. The paper is not well-organized. I focus here only on some points, which are hopefully easy for the authors to take into account in the revision.

1. Abstract-Line 24, GDP (gross domestic product)? Maybe gross domestic product (GDP)? Please confirm it.

2. Figure 1, is not clear enough. Please added national boundaries to make it clearer.

3. Line 143, CO2, please modify it and check the manuscript carefully.

4. Discussion looks good. However, the author did not discuss the limitation of this study. Furthermore, some important relevant references should be cited as follow.

1) The impact of urban renewal on land surface temperature changes: A case study in the main city of Guangzhou, China. Remote Sensing (2020), https://doi.org/10.3390/rs12050794.

2) Modelling spatial distribution of fine-scale populations based on residential properties, International Journal of Remote Sensing (2019), doi: https://doi.org/10.1080/01431161.2019.1579387.

3) Spatial Evolution of Population Change in Northeast China During 1992-2018, Science of the Total Environment (2021), https://doi.org/10.1016/j.scitotenv.2021.146023.

4) Contribution of urban ventilation to the thermal environment and urban energy demand: Different climate background perspectives, Science of the Total Environment (2021), https://doi.org/10.1016/j.scitotenv.2021.148791.

5) Suitability of human settlements in mountainous areas from the perspective of ventilation: a case study of the main urban area of Chongqing, Journal of Cleaner Production(2021), https://doi.org/10.1016/j.jclepro.2021.127467.

6) Optimizing local climate zones to mitigate urban heat island effect in human settlements, Journal of Cleaner Production (2020), https://doi.org/10.1016/j.jclepro.2020.123767.

7) Influence of urban morphological characteristics on thermal environment, Sustainable Cities and Society (2021), https://doi.org/10.1016/j.scs.2021.103045.

Reviewers' comments:

Reviewer's Responses to Questions

**Comments to the Author**

1. If the authors have adequately addressed your comments raised in a previous round of review and you feel that this manuscript is now acceptable for publication, you may indicate that here to bypass the “Comments to the Author” section, enter your conflict of interest statement in the “Confidential to Editor” section, and submit your "Accept" recommendation.

Reviewer #1: All comments have been addressed

Reviewer #2: All comments have been addressed

2. Is the manuscript technically sound, and do the data support the conclusions?

Reviewer #1: Yes

Reviewer #2: Yes

3. Has the statistical analysis been performed appropriately and rigorously? 

Reviewer #1: Yes

Reviewer #2: Yes

4. Have the authors made all data underlying the findings in their manuscript fully available?

Reviewer #1: Yes

Reviewer #2: Yes

5. Is the manuscript presented in an intelligible fashion and written in standard English?

Reviewer #1: Yes

Reviewer #2: Yes

6. Review Comments to the Author

Reviewer #1: Dear authors,

Thanks very much for your response letter. The issues that I worried about this manuscript are well explained, and detailed results are also provided. I recommend to publish this manuscript.

Reviewer #2: The authors explored the relationship between night-time light and socioeconomic factors. The research methodologies are reasonable, and the findings are justifiable. However, there are still a few aspects that should be improved to make the paper publishable. The paper is not well-organized. I focus here only on some points, which are hopefully easy for the authors to take into account in the revision.

1.Abstract-Line 24, GDP (gross domestic product)? Maybe gross domestic product (GDP)? Please confirm it.

2.Figure 1, is not clear enough. Please added national boundaries to make it clearer.

3.Line 143, CO2, please modify it and check the manuscript carefully.

4.Discussion looks good. However, the author did not discuss the limitation of this study. Furthermore, some important relevant references should be cited as follow.

1)The impact of urban renewal on land surface temperature changes: A case study in the main city of Guangzhou, China. Remote Sensing (2020), https://doi.org/10.3390/rs12050794.

2)Modelling spatial distribution of fine-scale populations based on residential properties, International Journal of Remote Sensing (2019), doi: https://doi.org/10.1080/01431161.2019.1579387.

3)Spatial Evolution of Population Change in Northeast China During 1992-2018, Science of the Total Environment (2021), https://doi.org/10.1016/j.scitotenv.2021.146023.

4)Contribution of urban ventilation to the thermal environment and urban energy demand: Different climate background perspectives, Science of the Total Environment (2021), https://doi.org/10.1016/j.scitotenv.2021.148791.

5)Suitability of human settlements in mountainous areas from the perspective of ventilation: a case study of the main urban area of Chongqing, Journal of Cleaner Production(2021), https://doi.org/10.1016/j.jclepro.2021.127467.

6)Optimizing local climate zones to mitigate urban heat island effect in human settlements, Journal of Cleaner Production (2020), https://doi.org/10.1016/j.jclepro.2020.123767.

7)Influence of urban morphological characteristics on thermal environment, Sustainable Cities and Society (2021), https://doi.org/10.1016/j.scs.2021.103045.

7. PLOS authors have the option to publish the peer review history of their article (what does this mean?). If published, this will include your full peer review and any attached files.

Reviewer #1: **Yes: **Yunxiang GUO

Reviewer #2: No

---

## [Author Response · Author response to Decision Letter 1]

1 Dec 2021

Dear Reviewer:

Thank you very much for your careful review of my manuscript, and points out some omissions in the manuscript. This has greatly helped this manuscript to have a chance to become a paper. We have carefully considered your all opinions, and made some changes in manuscript. The revisions use the "Track Changes" in the revised manuscript. Detail of the revisions are listed as follows.

1.Abstract-Line 24, GDP (gross domestic product)? Maybe gross domestic product (GDP)? Please confirm it

Thank you for your careful inspection. We have corrected this mistake in the manuscript.

2. Figure 1, is not clear enough. Please added national boundaries to make it clearer.

Thank you for your suggestion. We have added national boundaries to Figure 1 in our manuscript.

3. Line 143, CO2, please modify it and check the manuscript carefully.

I am sorry for my carelessness. We have checked all CO2 in the manuscript and changed it to CO2.

4. Discussion looks good. However, the author did not discuss the limitation of this study. Furthermore, some important relevant references should be cited as follow.

Thank you for your approval of the discussion section. In fact, we also talked about some limitations of the paper in the discussion and conclusion, such as the lack of research on spatial aggregation, factors, models and other issues, but they are scattered in the text and may be imperceptible. Now at the end of the discussion, we have added some new understanding of the limitations of this article. 

The references you recommended are very valuable, especially the articles on the spatial distribution and time evolution of population and buildings. In fact, there is also a discussion on this issue in the manuscript. We have added these references such as 50 and 54. Several other articles about urban thermal effect are not directly involved in this article, so we have not quoted them.

 If there is any question in the revised version of our manuscript, please feel free to contact us and we will make our best effort to meet your requirements. We sincerely thank you for consideration of our paper for publication in PLOS ONE.

Sincerely yours,

HAN Guhuai

---

## [Decision Letter · Decision Letter 2]

6 Dec 2021

PONE-D-21-17224R2The relationship between night-time light and socioeconomic factors in China and IndiaPLOS ONE

Dear Dr. Sun,

Thank you for submitting your manuscript to PLOS ONE. After careful consideration, we feel that it has merit but does not fully meet PLOS ONE’s publication criteria as it currently stands. Therefore, we invite you to submit a revised version of the manuscript that addresses the points raised during the review process.

We look forward to receiving your revised manuscript.

Kind regards,

Jun Yang

Academic Editor

PLOS ONE

Journal Requirements:

Additional Editor Comments:

I read this paper carefully. All my concerns have been addressed. However, there are many some points that need to be modified. Importantly, please check this paper.

(1) I found that the author order has changed, please explain it.

(2) Table 1 mentioned GDP and population, and it was derived from RESDC and RBI. The resolution of GDP and population in China is 1km, what is the resolution of it in India? In other words, if raster data is not needed, why not use statistics data?

(3) There are many mistakes, please check it carefully.

Line 160 - First, we converted the carbon emission NC format into TIF format with GDAL ‘.’ is missing.

Table title is missing, e.g. Line 254, 293 and 296.

Line 268 - R2of ?

(4) Fig 2 is incomplete, please modify it.

(5) Fig 3 - p value should be added in it. Please check it.

(6) Line 237 and 191 - where adjusted R2? Please check it.

Reviewers' comments:

Reviewer's Responses to Questions

**Comments to the Author**

1. If the authors have adequately addressed your comments raised in a previous round of review and you feel that this manuscript is now acceptable for publication, you may indicate that here to bypass the “Comments to the Author” section, enter your conflict of interest statement in the “Confidential to Editor” section, and submit your "Accept" recommendation.

Reviewer #1: All comments have been addressed

Reviewer #2: All comments have been addressed

2. Is the manuscript technically sound, and do the data support the conclusions?

Reviewer #1: Yes

Reviewer #2: Yes

3. Has the statistical analysis been performed appropriately and rigorously? 

Reviewer #1: Yes

Reviewer #2: Yes

4. Have the authors made all data underlying the findings in their manuscript fully available?

Reviewer #1: Yes

Reviewer #2: Yes

5. Is the manuscript presented in an intelligible fashion and written in standard English?

Reviewer #1: Yes

Reviewer #2: Yes

6. Review Comments to the Author

Reviewer #1: The author has carefully revised the article and my concerns have been well explained. I recommend to publish this paper.

Reviewer #2: I read this paper carefully. All my concerns have been addressed. However, there are many some points that need to be modified. Importantly, please check this paper.

(1) I found that the author order has changed, please explain it.

(2) Table 1 mentioned GDP and population, and it was derived from RESDC and RBI. The resolution of GDP and population in China is 1km, what is the resolution of it in India? In other words, if raster data is not needed, why not use statistics data?

(3) There are many mistakes, please check it carefully.

Line 160 - First, we converted the carbon emission NC format into TIF format with GDAL ‘.’ is missing.

Table title is missing, e.g. Line 254, 293 and 296.

Line 268 - R2of ?

(4) Fig 2 is incomplete, please modify it.

(5) Fig 3 - p value should be added in it. Please check it.

(6) Line 237 and 191 - where adjusted R2? Please check it.

7. PLOS authors have the option to publish the peer review history of their article (what does this mean?). If published, this will include your full peer review and any attached files.

Reviewer #1: No

Reviewer #2: No

---

## [Author Response · Author response to Decision Letter 2]

22 Dec 2021

Dear Reviewer:

Thank you very much for your careful review of my manuscript, and points out some omissions in the manuscript. This has greatly helped this manuscript to have a chance to become a paper. We have carefully considered your all opinions, and made some changes in manuscript. The revisions use the "Track Changes" in the revised manuscript. Detail of the revisions are listed as follows.

1. I found that the author order has changed, please explain it.

All the listed authors have made great contributions to this paper. The order of authors was adjusted a long time ago, which was decided by contributions and funds. The last manuscript we revised didn't change the order of authors, only Dr. Sun's author affiliation. Dr. Sun is a doctoral student who graduated from Peking University and is now teaching at Dalian Maritime University. So we added his author affiliation 3.

2. Table 1 mentioned GDP and population, and it was derived from RESDC and RBI. The resolution of GDP and population in China is 1km, what is the resolution of it in India? In other words, if raster data is not needed, why not use statistics data?

Thank you very much for your careful consideration of our statistical process, this problem is actually a difficulty that we have faced in the statistical process. 

We only processed the state-level data in India (compared with the provincial level in China), so we directly used the statistical table data divided by administrative region. For China, in addition to the provincial level, we also process county-level data, so the resolution of statistical data is required to be higher. There are more than 2,800 county-level administrative regions in China, but China's Bureau of Statistics does not provide a unified data table, so it would be a lot of work to search each county. However, the Resource and Environmental Science Data Center of Chinese Academy of Sciences(www.resdc.cn) has provided 1km grid data of GDP and population, and this raster data is also made according to the county-level and township-level data of Chinese statistics, which is authoritative. Therefore, it is much more convenient to use raster data in China, and it is also conducive to some possible subsequent spatial relationship analysis or more detailed work.

3. There are many mistakes, please check it carefully.

Line 160 - First, we converted the carbon emission NC format into TIF format with GDAL ‘.’ is missing.

Table title is missing, e.g. Line 254, 293 and 296.

Line 268 - R2of ?

I am sorry for our carelessness. We have changed the problem you mentioned, and checked the similar problems in the full text. Since the two tables belong to the same multiple regression, I added subheadings to them. Line 268 - R2 are various regression R2of India (Fig 7). 

4. Fig 2 is incomplete, please modify it.

Thank you for your suggestion. We have modified Figure 2 to make it correspond to the method in the paper. 

5. Fig 3 - p value should be added in it. Please check it.

Thank you for your reminder. The analysis of this article mainly focuses on regression coefficient, R2 and MRE, so only regression coefficient, R2 and MRE are put in all the figures. We think that P value or F value can only be mentioned in the text or table to represent the persuasiveness of the data. After your reminder, we found that the p value was wrongly written when copied from SPSS to the text. It should be less than 0.01 instead of 0.00. We have made a comprehensive revision.

6. Line 237 and 191 - where adjusted R2? Please check it.

In multiple regression, there will be adjusted R2 affected by the number of dependent variables. Therefore, in order to prevent the negative effects of incorrect dependent variables, we recorded R2 and adjusted R2 for comparison. There are some punctuation errors here, and we have made unified changes.

 If there is any question in the revised version of our manuscript, please feel free to contact us and we will make our best effort to meet your requirements. We sincerely thank you for consideration of our paper for publication in PLOS ONE.

Sincerely yours,

HAN Guhuai

---

## [Decision Letter · Decision Letter 3]

27 Dec 2021

The relationship between night-time light and socioeconomic factors in China and India

PONE-D-21-17224R3

Dear Dr. Sun,

We’re pleased to inform you that your manuscript has been judged scientifically suitable for publication and will be formally accepted for publication once it meets all outstanding technical requirements.

Kind regards,

Jun Yang

Academic Editor

PLOS ONE

Additional Editor Comments (optional):

Accept

Reviewers' comments:

Reviewer's Responses to Questions

**Comments to the Author**

1. If the authors have adequately addressed your comments raised in a previous round of review and you feel that this manuscript is now acceptable for publication, you may indicate that here to bypass the “Comments to the Author” section, enter your conflict of interest statement in the “Confidential to Editor” section, and submit your "Accept" recommendation.

Reviewer #2: All comments have been addressed

2. Is the manuscript technically sound, and do the data support the conclusions?

Reviewer #2: Yes

3. Has the statistical analysis been performed appropriately and rigorously? 

Reviewer #2: Yes

4. Have the authors made all data underlying the findings in their manuscript fully available?

Reviewer #2: Yes

5. Is the manuscript presented in an intelligible fashion and written in standard English?

Reviewer #2: Yes

6. Review Comments to the Author

Reviewer #2: (No Response)

7. PLOS authors have the option to publish the peer review history of their article (what does this mean?). If published, this will include your full peer review and any attached files.

Reviewer #2: No

---

## [Editor Report · Acceptance letter]

5 Jan 2022

PONE-D-21-17224R3 

The relationship between night-time light and socioeconomic factors in China and India 

Dear Dr. Sun:

I'm pleased to inform you that your manuscript has been deemed suitable for publication in PLOS ONE. Congratulations! Your manuscript is now with our production department. 

Kind regards, 

on behalf of

Dr. Jun Yang 

Academic Editor

PLOS ONE